# Influence of Farm Management for Calves on Growth Performance and Meat Quality Traits Duration Fattening of Simmental Bulls and Heifers

**DOI:** 10.3390/ani9110941

**Published:** 2019-11-09

**Authors:** Denis Kučević, Tamara Papović, Vladimir Tomović, Miroslav Plavšić, Igor Jajić, Saša Krstović, Dragan Stanojević

**Affiliations:** 1Faculty of Agriculture, University of Novi Sad, Trg D. Obradovića 8, 21000 Novi Sad, Serbia; denis.kucevic@stocarstvo.edu.rs (D.K.); plavsic@gmail.com (M.P.); igor.jajic@stocarstvo.edu.rs (I.J.); sasa.krstovic@stocarstvo.edu.rs (S.K.); 2Faculty of Technology, University of Novi Sad, Bulevar cara Lazara 1, 21000 Novi Sad, Serbia; tomovic@uns.ac.rs; 3Faculty of Agriculture, University of Belgrade, Nemanjina 6, 11080 Beograd, Zemun, Serbia; stanojevic@agrif.bg.ac.rs

**Keywords:** heifer, bull, Simmental, fattening, management, carcass and meat quality

## Abstract

**Simple Summary:**

Cattle have been selected for their adaptation to a specific environment and productive system, in which they show, in theory, their best economical results. With appropriate nutrition, the calf’s performance enhances during early life and improve the production limit providing distinctive opportunities to optimize feeding strategies and increase the profitability of beef production. There is considerable variation in fattening protocols as well as in farm conditions. Meat quality parameters and carcass traits are the main objectives of most research carried out in the beef production area. Optimizing meat quality parameters and carcass traits are important for farmer profits and consumer satisfaction. According to that, at the phenotypic level, growth performance and traits could be observed. Rearing practices are known to have an impact on cattle carcasses and meat characteristics. The rearing practices applied after calving have an influence on the animal’s performance at the growth period and can involve different animal properties at the beginning of the fattening period.

**Abstract:**

This study assessed the effects of farm management during rearing practices in the first months of a calf’s life on growth performance and meat quality traits during the fattening period. A total of 48 Simmental calves were divided into two groups at a commercial cattle feedlot. In the first group were calves from the same farm and herd (n = 12 male and n = 12 female). The second group included calves from several different herds and farms (n = 12 male and n= 12 female). Calves were transferred to a feedlot and fed with a commercial feedlot ration at three to four months of age. The aim was to determine if identical fattening conditions at feedlot can reduce initial calf rearing differences between cattle during the fattening period. Bulls grew faster than heifers reaching higher total gain and showed significantly higher slaughter weight than heifers. Meat samples of heifers from the same herd had the highest intramuscular fat content and reddest color with significant differences among cattle groups. The most abundant fatty acid was oleic acid (C18:1), followed by palmitic (C16:0), stearic (C18:0), linoleic (C18:2), and myristic acid (C14:0). Meat samples of heifers from different herds were darkest with highest content of iron (Fe) with significant differences among cattle groups.

## 1. Introduction

There is wide variation in meat production and productivity levels. Variations in these production traits can be attributed to differences in genetic composition, nutrition, slaughter endpoints, and gender [1,2]. The bulls grow faster and more efficiently, had a higher slaughtering proportion, and produce leaner carcasses with a higher proportion of total meat than heifers. Therefore, the meat from heifers compared to bulls have more dry matter and intramuscular fat, and is more tender and acceptable [3,4]. Many studies showed that different rearing factors applied during the fattening period have an impact on carcass or meat properties [4,5,6]. Further, it has been shown that rearing management before the fattening period could impact both carcass [7,8,9] and meat quality traits. Hence, the consideration of a wider period rather than the fattening period alone could be of great interest to improve the prediction power of carcass and meat quality traits. There is considerable variation in fattening protocols as well as in farm conditions [10]. The rearing practices applied after calving have an influence on the animal’s performance at the growth period. These differences in performance involve different animal properties at the beginning of the fattening period [11]. 

Constant dynamic changes in industry demand experts with multidisciplinary knowledge and skills with the need to find faults in the production processes in a short time but also to react preventively in order to enable continual process workflow [12]. Currently developed cattle identification systems are based on electronic technologies that allow automation, instead of traditional systems based on visual identification [13]. An automated system can work autonomously, and, if required, can be easily integrated into the new or existing complex farm management system [13] and also improve consumer confidence and provide assurance to buyers regarding the animal’s life history [14]. According to topics of interest, developers of new products and services need to do thorough analysis of information available in patent databases and to use collected information for defining future research and development plans and market strategies [15]. Producing a product that delivers a consistently high-quality eating experience is paramount to the beef industry to ensure consumer satisfaction [16].

Simmental cattle, a dual purpose worldwide breed common in central Europe, is usually slaughtered between 16–18 months and 600–700 kg live weight [17]. Considering that Simmental is the most widespread breed in Serbia (more than 70%) and because of the agro-climatic conditions, intensive systems of fattening based on concentrates ad libitum and cereal straw, with young animals, are the most common type of beef production systems. Calves from intensive systems are housed indoors, weaned at an early age (two to four months) and reared with concentrate and cereal straw ad libitum, when their diet is switched to concentrate [18].

Male and female calves from different farms for this research were considered together during the fattening period under identical conditions and also their expression of the observed parameters at the phenotypic level. We hypothesized that the different rearing practices from the first three to four months of calf’s life can influence the characteristics of the beef carcasses and quality of produced meat. Moreover, differences between Simmental bulls and heifers in relation to growth performance, carcasses, and meat quality traits were significant and in agreement with our expectations that calves from the identical rearing conditions have more similar final results and with those reported in the literature. 

## 2. Materials and Methods 

### 2.1. Animals and Growth Performance, Slaughter Procedures, and Carcass Quality

The investigation was conducted on 48 calves of Simmental breed produced under an intensive rearing system at commercial beef feedlot. A total of 24 calves came from the same herd (from one farm 12 male and 12 female) from intensive system. They were weaned early and started with four weeks of age to be fed with concentrate (corn middlings 43%; limestone flour 25%; sunflower meal 19%; soybean meal 10%; premix 1.5%; limestone 1%; monocalcium phosphate 1%; animal feed salt 0.5%), and oats straw. The other 24 calves were from several different herds (from different farms including the same number of male and female) reared in semi-intensive system with different rearing practices. For the fattening period, the two groups were housed at the commercial beef feedlot. Calves from the same herd previously carried out as the first group and the second group included calves from several other herds. They all were up from three to four months of age when transferred to feedlot and fed a commercial feedlot ration. The adaption period was three weeks. During that, animals started to consume ad libitum the same diet and reared under the same environmental and production regime. The fattening period ends when bulls reached up 568 to 613 kg and heifers reached up 517 to 547 kg of body weight.

During the fattening period, the rearing system was free, and food consisted of concentrated feeds, hay and corn grain silage locally produced and were formulated to meet the nutrient requirements [18] for the different growth phases. Animals had ad libitum access to water during the whole fattening period. Changing the concentrate composition at body weight of 250–300 kg (from all the way through to and finish phase of fattening) was a correction associated with declining ratio of protein to energy connected with age. The cattle were fed ad libitum a total mixed ration (TMR) composed of corn grain and maize silage (70%) and concentrate (30% in total, including: Corn middlings 4.3%; sunflower meal 70%; limestone flour 15%; premix 3%; limestone 3%; monocalcium phosphate 3%; animal feed salt 1.5%).

Data for each animal included initial weight (kg), total gain (kg), slaughter weight (kg), fattening period (days), and slaughter age (days) which were recorded systematically. Individual calves weights were measured using a heavy duty scale with accuracy ± 0.5 kg (initial weight) at the beginning and the end of the fattening period prior to slaughter. An estimated total gain during the fattening period was calculated between the initial weight and at slaughter weight. When the target slaughter age was achieved, the cattle were slaughtered in the slaughterhouse. 

### 2.2. Slaughter Procedures

From feedlot to slaughterhouse, cattle were transported unmixed in early morning hours and after transport, which took about 3 h (farms are 60 km far from the slaughterhouse), animals were rested for about 2 h in the abattoir. The animals were rested by isolating them from noise and human activity during the lairage period. All the cattle were slaughtered according to routine procedures of the slaughterhouse. Carcasses were conventionally chilled for 24 h in a chiller at 0–4 °C. After chilling, *M. longissimus lumborum* (LL) was removed from the right side of each carcass, in the area between the sixth and seventh rib to determine meat quality. The meat samples were trimmed of visible adipose and connective tissue. Physical and sensory characteristics were measured on fresh or cooked beef. Samples for chemical analysis (approximately 250 g) were taken after the homogenization of the LL; they were vacuum packaged in polyethylene bags and stored at −40 °C until analysis.

### 2.3. Carcass Quality Traits Evaluation

The carcass quality was characterized by: Hot carcass weight (HCW), dressing percentage (ratio between hot carcass weight and live weight before slaughter, in %), and conformation score. Carcass conformation was graded under the EU beef carcass classification (SEUROP) scheme. After slaughtering of the animals, their carcass weighing and muscle development evaluation was done [19]. Beef carcass conformations are defined with the EUROP scale, represented by the letters E, U, R, O, and P (class S is used only in countries where there is a basis for its use—double muscled cattle). The scoring consists of a visual assessment of carcass muscling where carcasses graded as E have the most muscularity, and this decreases through to P which have the least muscularity (muscle development). At the same time, the degree of fat cover of the carcasses was based on visual evaluation numerically scored from 1 = very low to 5 = very high, according to the same European classification [19].

### 2.4. Meat Quality Measurements

#### 2.4.1. Physical and Sensory Quality Measurements

The pH value was measured in the center of LL muscles at 24 h (pH_24_ h) *post-mortem* [20,21]. *S*amples for color measurements were taken from the central part of all muscles, perpendicularly to the long axis of LL, after 60 min of blooming [22]; the minimum thickness of samples was 2.5 cm. The instrumental color was determined using Minolta Chroma Meter CR-400 (Minolta Co., Ltd., Osaka, Japan) using D-65 lighting, a 2° standard observer angle and an 8 mm aperture in the measuring head. The CIE *L*a*b** color coordinates [23] were lightness (*L**), redness (*a**), yellowness (*b**), *C** (chroma—saturation index; *C** = (*a**2 + *b**2)1/2), *h* (hue angle; *h* = arctangent (*b**/*a**)), and *λ* (dominant wavelength (nm)) [23,24,25]. Water-holding capacity (WHC) was determined as free water (exudative juice) using the filter paper press method [21,26,27]. The cooking loss was determined by the method as described by Tomović et al. [28]. Samples of cooked meat, after cooking loss determination, were used for objective determination of tenderness [28,29]. Tenderness was measured as the shear force (N) using Warner–Bratzler shear machine (Model SD—50 of 50 lb or 222 N capacity, John Chatillon & Sons, New York, NY, USA) as described by Senk et al. [12]. The sensory analyses were performed by an eight-member panel. Samples for sensory evaluation were taken perpendicularly to the long axis of LL; the minimum thickness was 2.54 cm. Panelists evaluated color using sets of [25] official color (1 = extremely bright cherry-red to 8 = extremely dark red) and marbling [30] (1 = slight to 7 = moderately abundant) standards.

#### 2.4.2. Proximate and Mineral Composition 

Moisture [31], protein (nitrogen × 6.25; [32]), total fat [33], and total ash [34] contents of muscles were determined according to methods recommended by the International Organization for Standardization. The minerals contents of the meat (calcium (Ca), sodium (Na), magnesium (Mg), iron (Fe), zinc (Zn), and copper (Cu)) were determined by the flame atomic absorption spectrometry as described in detail described by Tomović et al. [35] after mineralization by dry ashing [34]. Phosphorus (P) was determined by the standard spectrophotometric method [36]. All analyses were performed in duplicate.

#### 2.4.3. Fatty Acids Composition 

Meat samples of 5 g were dried at the temperature of 105 °C. Then, samples were quantitatively transferred into an extraction cartridge, and petroleum ether extraction was run for 5 h in the Soxhlet extractor [37,38]. The methyl esters of the fat extracted were formed according to the method described by Yurchenko et al. [39]. Fatty acids methyl esters were identified by comparing the retention times of fatty acid methyl ester peaks from samples with those of standards obtained from Supelco (Supelco C4-C24 Even Carbon for saturated: C14, C16, C18, and Supelco Fame Mix GLC-10 for unsaturated fatty acids: C18:1, C18:2). Chromatographic analysis of the methyl esters was carried out with a gas chromatograph GC-2010 Plus, Shimadzu, equipped with a flame ionization detector and autosampler AOC-20i, Capillary Column InterCap WAX (length 30 m, inner diameter 0.25 mm, film thickness 0.25 μm). Analysis of the standard mixture of methyl esters was carried out using reference probe sample of 0.6 μL at split ratio 40:1. The injector and detector temperatures were 260 °C, and the analysis was performed in isothermal conditions at 200 °C. Helium was applied as carrier gas with flow rate of 3 mL/min.

### 2.5. Statistical Analysis

All data are presented as mean and standard deviation (SD). Data were studied by two-way factorial ANOVA (gender and group) and Post-Hoc test (Duncan’s multiple range test) was used to characterize statistically significant differences at the level *p* < 0.05 between analyzed groups within the Statistica software package (ver. 13 StatSoft, Inc. 2016, Kraków, Poland). The two-way factorial model equation used for the evaluation was as follows:Y_ijkl_ = μ + F_i_ + G_j_ + I_k_ + e_ijk_(1)
where: Y_ijkl_, the value of the tested traits (dependent variable); µ, average mean value of the dependent variable; F_i_, fixed effect of the group (i = 1,2); G_j_, fixed effect of the gender (j = 1,2); I_k_, interaction group x gender; e_ijk_, other random effects. 

## 3. Results and Discussion 

### 3.1. Growth Performance

In this study, weights were recorded at the beginning and at the end of the fattening period. Despite the fact that calves came from different herds, there were no significant differences between two groups in initial weight at the start of the fattening period which is presented in Table 1. The average of days spent in a feedlot for the second group of cattle from different herds was significantly longer compared to the first group of cattle which were from the same herd. It is well known that the optimal slaughter ages and weights vary widely among cattle breed types depending on how rapidly they mature, which is characterized by fat deposition during the “finishing” period [1]. In this research, the group had significant influence at the slaughter age (*p* < 0.001). Considering slaughter age, the bulls and heifers from the second group were older (512.2 and 530.3 days, respectively) than those from the first group (491.6 days). Moreover, cattle from the second group spent a longer period in the feedlot, which can be explained by the fact that the calves from different herds brought to the same feedlot took a longer period to adapt at the beginning, especially the females. 

There was interaction between gender and group for total gain during the fattening period (*p* < 0.001), with bulls achieving higher total gain than heifers. In our study, heifers from the second group achieved lowest total gains and slaughter weight (383 and 518 kg) during the fattening period in comparison with the rest of the animals. Likewise, heifers from the second group spent the longest period at the feedlot (456.8 days) which corresponded to the above-mentioned claim that calves from different herds with different rearing practices should take a longer period to adapt. 

A higher total gain of bulls compared to heifers here resulted in higher slaughter weight of bulls. Similarly, Bureš et al. [3] found a higher slaughter weight for bulls compared to heifers 18 months old, fattened in quite identical husbandry conditions. These results are in accordance with data obtained by Kaminiecki et al. [40] for Charolais x Simmental crossbreeds bulls. 

### 3.2. Carcass Quality Traits Evaluation

The carcass quality traits of cattle are shown in Table 2. Hot carcass weights from bulls (354 and 379 kg) were significantly higher than from heifers (327 and 309 kg) in the first and second group, respectively (*p* < 0.001). Our results for the carcass weight were lower than those [41] published for Simmental bulls and higher for the dressing percentage [1,17]. The effect of nutrition efficiency increased with slaughter weight due to the interaction between the total gain during the fattening period and the slaughter weight which resulted in higher values of the carcass weight and dressing percentage. Moreover, Herva et al. [42] concluded that carcass fat content was increased when carcasses were heavier, and when a daily gain was higher.

Group, gender, and their interaction did not significantly affect the dressing and conformation traits evaluation (*p* > 0.27). Kaminiecki et al. [40] found that Simmental × Charolais crossbreeds produced a dressing percentage of 58.5% while [43] reported that carcass dressing percentage was higher in heavier animals, which could result from higher carcass fatness. Both studies were in accordance with our results. Higher final weights of bulls in our trial resulted higher hot carcass weight compared to heifers, however the dressing percentage was not affected. Fat cover scores were significantly influenced by gender. Usually females start to deposit fat earlier than males. In addition, the males were intact (with their testicles), so they should be leaner than heifers. The results regarding fat cover evaluated indicate that most animals belonged between score three and four. Regarding conformation, the majority of cattle carcasses were classified as class R. Bulls showed significantly higher scores of fat cover (4) than heifers (3.8 and 3.6). Our results for bulls were in accordance with results obtained by Chambaz et al. [44] for Simmental steers (conformation score 3.7 which present U class and fatness score 4.1). According to Monteils et al. [8] irrespective of a cattle category, the higher carcass conformation and higher carcass fat cover were found related to increased hot dressing percentage. Interestingly, in each analyzed cattle group in our research, among the carcasses classified to a higher slaughter weight, a higher grade (conformation, fat cover) was recorded.

### 3.3. Physical and Sensory Quality Measurements

The data of the Simmental cattle showed variations in the properties of interest referring to physical and sensory traits depending on the examined effects (Table 3). In the present study, pH_24_ value was significantly influenced by the gender, but all mean pH_24_ values fell in a very narrow range with 5.44 (heifers) to 5.50 (bulls) which was in accordance with the results obtained by Pilarcyk [45] for Simmental bulls (pH_24_ 5.52). Meat of high quality has pH at the range of 5.4–5.6, but meat of a higher pH value can be characterized by gummy structure, increased water-holding capacity, and decreased specific taste [4]. We found that an interaction effect between the group and gender was found for all instrumental color parameters (*p* < 0.001). All instrumental color parameters showed significant differences between cattle. Significantly paler (lightest color, higher *L**) numerical CIE*L** mean values were found in meat samples from bulls on the second group (39.76) and the lowest (darkest color) was found in meat samples from heifers at the same group (37.73).

Furthermore, meat of heifers from the first group had the reddest color (CIE*a** value was 22.13). As well, heifers from the first group also had the significantly highest CIE*b** value (10.16). Brighter color of meat from heifers as compared with meat from bulls could be due to the increased fat disposition content of heifers as fat increases brightness of meat color and fiber type as well [4]. Concomitant, heifers from the first group had significantly highest values of CIE*C** (24.36). Bulls from the second group had significantly higher value for the *h* (hue angle) (25.42) and the lower value of *λ* (dominant wavelength) (607.60 nm) than the rest of the animals. A lower *L** value and yellowness *b** were found in the meat of older cattle (heifers from the second group) whereas hue angle (*h*) was similar for all animals, which was in accordance with [46].

WHC (M/T cooking losses) was influenced by the group. Cattle from the second group had better WHC (M/T = 0.41 for bulls and 0.39 for heifers, a bigger M/T ratio indicating a better WHC) than cattle from the first group (M/T = 0.36, for both). If more water is retained in the muscle/myofibrillar structure, generally a product with a higher sensory tenderness and juiciness is obtained [14].

Gender significantly affected the cooking loss (*p* < 0.001). Bulls showed higher content of cooking loss (38.34% and 37.17%) than heifers (33.93% and 33.30%). Moreover, cattle from the first group had higher content than cattle from the second group comparing in total. Values for cooking loss in our study were similar to the results of Scollan et al. [41] for crossbred Charolais × White Holstein-Friesian bulls (34.53%). Significant effect of gender and interaction between group and gender was found for WBSF. Bulls showed significantly higher WBSF value (56.03 and 61.02 N) than heifers (52.98 and 50.13 N) for the first and second group, respectively. Weglarz [4] found that comparing meat from bulls and heifers, heifer meat appeared slightly more tender, which must have been related to the higher content of intramuscular fat. A slightly lower Warner–Blatzer shear force values than those in our study, were reported by Bureš and Barton [47] for Fleckvieh bulls (49.8 N) and for Simmental bulls (48.19 N) [48]. Beef from cattle with a high intramuscular fat level often has a lower shear force [49], which is in accordance with results from our study. The color sensory attribute of the meat samples did not differ significantly between the cattle groups (*p* > 0.05). Scollan et al. [41] demonstrated that the meat from lighter and younger animals was significantly more tender, however with larger variation within WBSF values. Marbling score is being used as an indirect mean for meat sensory quality assessment [50]. There was an interaction effect between group and gender for marbling score. Marbling score was significantly highest (*p* < 0.05) for the heifers at the second group (4.17) than the other animals. 

### 3.4. Proximate and Mineral Composition

The proximate composition of meat samples from Simmental cattle are shown in Table 4. We found an interaction effect between the group and gender for moisture content (*p* < 0.001) where the bulls had higher moisture contents (73.21% to 74.54%) than heifers (up 72.11% to 72.24%) for the first and second group, respectively. Proximate composition, except protein was influenced mainly by the gender. No differences were found in the content of protein among meat samples from two groups. As expected, the protein content was in agreement with some earlier investigations [39,43].

Gender significantly affected (*p* < 0.001) content of total fat and total ash. However, the content of total fat was the most variable inside the investigated groups. Total fat content was significantly higher for heifers (ranged between 5.19% to 5.40%) and lower for bulls (3.00% to 4.38%) at the first and second group, respectively. According to the results of Weglarz [10] that are comparable to ours, meat from bulls had higher moisture and significantly lower fat and total ash content in comparison with meat from heifers. Content of total ash was significantly higher for heifers (1.13%) than for bulls (from 1.04% to 1.08%) which was in accordance with total ash content reported by Pilarczyk et al. [45] and Monteils et al. [43].

An overview of obtained results for the mineral composition of meat samples are presented in Table 5. Gender affected the content of phosphorus, calcium, iron, and zinc in the meat samples (*p* < 0.001). Phosphorous was the most abundant mineral in fresh meat samples. As shown in Table 6, the content of phosphorous was significantly higher for the bulls (152.28 and 157.97 mg/100g) than for heifers (106.91 and 110.26 mg/100g) from the first and second group, respectively. Accordingly, bulls showed significantly higher content of calcium than heifers. Interaction effect between group and gender was found to be significant for magnesium content. The highest magnesium content was found in the meat samples for bulls from the first group (24.61 mg/100g). All investigated effects (group, gender, and their interaction) significantly affected (*p* < 0.001) the content of iron and zinc. Heifers showed significantly higher content of iron compared to bulls with significant differences among cattle groups. Heifers from the second group had a significantly higher content of iron in meat samples (2.46 mg/100g) in regard to rest animals. According to Domaradzki et al. [50] a similar variation those to ours in the content of minerals of young Simmental bulls is reported. 

There were noticeable significant differences between investigated groups for the content of zinc, where the highest content of zinc was found for bulls from the first group (6.26 mg/100g) and the lowest for bulls from the second group (5.21 mg/100g). Investigated effects did not significantly affect sodium and copper content in meat samples, and there were no differences between groups. Nogalski et al. [51] said that breed is a significant factor determining the content of minerals in the muscles of cattle raised under the same conditions. 

### 3.5. Fatty Acids Composition

The results for the fatty acid profile of meat samples for investigated groups are presented in Table 6. In general, the most abundant fatty acid was oleic acid (C18:1) with g/100g fat up 40.26 to 42.21 for bulls and 42.75 to 44.98 g/100g fat for heifers, followed by palmitic (C16:0), stearic (C18:0), linoleic (C18:2), and myristic acid (C14:0). Gender significantly affected oleic acid (C18:1) content (*p* < 0.001) where the heifers had significantly higher oleic acid (C18:1) content (44.98 and 42.75 g/100g fat) than bulls (42.21 and 40.26 g/100g fat). Gender also significantly affected stearic acid (C18:0) composition, where bulls had significantly higher content of stearic acid (C18:0) (19.28 and 18.50 g/100g fat) than heifers (16.37 and 17.06 g/100g fat). Results reported by Monteils et al. [43] for oleic acid (C18:1) and linoleic acid (C18:2) were lower compared to our results. In the research of [43], a higher IMF content of meat was associated with a considerable increase in MUFAs concentrations and a decrease in PUFAs levels which could result from feeding grass silage ad libitum. The interaction between the group and gender had significant influence on saturated fatty acids (*p* < 0.001), such as myristic acid (C14:0) and palmitic acid (C16:0). Content of myristic acid (C14:0) was significantly higher for bulls at the first group (2.56 g/100g fat) and significantly lower for the bulls at the same group (2.18 g/100g fat). A difference between investigated animals for palmitic acid (C16:0) content was significant. Similarly, the heifers from the second group had the highest palmitic acid (25.36 g/100g fat) content, while the lowest palmitic acid content was obtained for bulls from the same group (23.32 g/100g fat). Results obtained in studies by [6,43,52] were similar to ours.

De Smet et al. [53] found that an increased fat content of bovine meat was paralleled by increased proportions of SFAs and MUFAs, and a decreased proportion of PUFAs. Therefore, it is known that FA composition is mainly affected by rearing and feeding conditions. Feed composition is known to be one of the most important factors influencing fatty acids composition in beef. Some researchers [34,35] demonstrated that when animals were grown at the same rate, muscles from cattle which had a high grass intake had a higher PUFA/SFA ratio and a lower n-6/n-3 PUFA ratio in comparison with muscles from cattle fed concentrates. Cattle with a high potential for lean beef production are frequently fattened on concentrate diets, which may be unfavorable for the n-6/n-3 polyunsaturated fatty acids ratio in meat. The reason for this is the fact that the fat in concentrates contains higher levels of C18:2n-6. Introducing forage in the diet of beef cattle should enhance the n-3 fatty acid concentrations since forages are a good source of C18:3n-3 [43]. 

## 4. Conclusions

The results from this research suggested that a fattening period of around 400 days is more than sufficient to eliminate differences which can be caused by the different rearing system and farm management for calves before the fattening. Therefore, more uniform values for most of the examined traits were achieved within the first group where the cattle for the fattening were from the same herd. So, the group had significant effect on the age of slaughter, where the cattle from the second group spent significantly longer time there and were older than the cattle from the first group. This can be explained by the fact that the calves from different herds took a longer period to adapt, especially the female population. Rearing practices and the production system might modify some of the characteristics of commercial beef, especially those associated with fat. Moreover, variability of rearing factors could make difficulties to simultaneously analyze their impacts on the carcass and the meat. Slaughter traits such as quality of meat samples may vary depending on the combinations of rearing practices utilised. For the future investigation in addition, it would be necessary to collect breeding data on female and male cattle with more detailed rearing practices before the fattening period to refine the characterization of management system with a shorter period of fattening.

## Figures and Tables

**Table 1 animals-09-00941-t001:** Growth performance per groups and gender of Simmental cattle fattened in a feedlot.

Parameter	1st Group (Same Herd)		2nd Group (Different Herds)		*p*-Values		
	Male	Female	Male	Female	Group	Gender	Group × Gender
**IW (kg)**	142.4 ± 23.2	145.0 ± 9.4	148.3 ± 17.1	135.1 ± 11.6	0.672	0.262	0.099
**TG (kg)**	426.3 ^b^ ± 33.8	402.5 ^c^ ± 25.9	465.4 ^a^ ± 23.2	382.8 ^c^ ± 31.0	0.250	<0.001	<0.001
**SW (kg)**	568.8 ^b^ ± 35.3	547.5 ^b^ ± 24.8	613.8 ^a^ ± 31.78	517.9 ^c^ ± 24.2	0.369	<0.001	<0.001
**DIF (days)**	416.3 ^b^ ± 19.9	410.7 ^b^ ± 16.7	421.8 ^b^ ± 17.5	456.8 ^a^ ± 12.9	<0.001	<0.001	<0.001
**SA (days)**	491.6 ^b^ ± 27.6	491.7 ^b^ ± 14.5	512.2 ^ab^ ± 43.3	530.3 ^a^ ± 9.9	<0.001	0.253	0.257

IW = initial weight (at start of the fattening period); TG = total gain during the fattening period; SW = slaughter weight; DIF = days in feedlot; SA = slaughter age. ^a,b,c^ Row means with different superscript differ in significance at *p* < 0.05.

**Table 2 animals-09-00941-t002:** Carcass quality traits evaluation for investigated groups of Simmental cattle.

Parameter	1st Group (Calves from the Same Herd)		2nd Group (Calves from the Different Herds)		*p*-Values		
	Male	Female	Male	Female	Group	Gender	Group × Gender
**HCW (kg)**	354.0 ^b^ ± 18.5	327.9 ^c^ ± 16.7	379.4 ^a^ ± 23.4	309.7 ^c^ ± 39.1	0.634	<0.001	<0.001
**Dressing (%)**	62.3 ± 1.7	59.9 ± 1.5	61.8 ± 1.2	59.7 ± 6.2	0.726	0.271	0.877
**Conformation**	2.8 ± 0.4	2.6 ± 0.4	2.8 ± 0.4	2.5 ± 0.4	0.869	0.141	0.620
**Fat cover**	4.0 ^a^ ± 0.4	3.8 ^ab^ ± 0.2	4.0 ^a^ ± 0.2	3.6 ^b^ ± 0.4	0.394	<0.001	0.204

HCW = hot Carcass weight; Dressing = dressing percentage; Conformation = conformation scores, EUROP classification scales from E = 5 excellent; U = 4 very good; R = 3 good; O = 2 fair; P = 1 poor; Fat cover = fat cover scores, EUROP classification scales from 1 = low; 2 = slight; 3 = average; 4 = high and 5 = very high. ^a,b,c^ Row means with different superscript differ in significance at *p* < 0.05.

**Table 3 animals-09-00941-t003:** Physical and sensory quality measurements of fresh and cooked *M. longissimus lumborum* from investigated groups of Simmental cattle.

Parameter	1st Group (Same Herd)		2nd Group (Different Herds)		*p*-Values		
	Male	Female	Male	Female	Group	Gender	Group × Gender
**pH_24_**	5.50 ^a^ ± 0.04	5.45 ^ab^ ± 0.02	5.50 ^a^ ± 0.04	5.44 ^b^ ± 0.10	0.817	<0.001	0.817
***L****	38.22 ^bc^ ± 1.32	39.02 ^ab^ ± 1.57	39.76 ^a^ ± 1.89	37.73 ^c^ ± 1.04	0.780	0.157	<0.001
***a****	19.60 ^c^ ± 1.02	22.13 ^a^ ± 1.11	20.79 ^b^ ± 1.00	19.81 ^bc^ ± 2.01	0.158	0.054	<0.001
***b****	8.84 ^b^ ± 0.83	10.16 ^a^ ± 0.89	9.90 ^a^ ± 0.70	8.42 ^b^ ± 1.05	0.192	0.756	<0.001
***C****	21.51 ^c^ ± 1.23	24.36 ^a^ ± 1.37	23.03 ^b^ ± 1.16	21.53 ^c^ ± 2.23	0.158	0.141	<0.001
***h***	24.20 ^b^ ± 1.29	24.54 ^ab^ ± 0.86	25.42 ^a^ ± 0.94	22.93 ^c^ ± 1.13	0.523	<0.001	<0.001
***λ*** **(nm)**	609.16 ^b^ ± 1.90	609.34 ^b^ ± 1.08	607.60 ^c^ ± 1.35	611.47 ^a^ ± 1.97	0.546	<0.001	<0.001
**WHC-M (cm^2^)**	4.10 ^b^ ± 0.45	3.98 ^b^ ± 0.19	4.64 ^a^ ± 0.46	4.25 ^b^ ± 0.39	<0.001	<0.001	0.244
**WHC–T (cm^2^)**	11.37 ^a^ ± 0.40	11.15 ^ab^ ± 0.33	11.42 ^a^ ± 0.42	11.03 ^b^ ± 0.33	0.744	<0.001	0.432
**WHC–RZ (cm^2^)**	7.26 ^a^ ± 0.59	7.18 ^ab^ ± 0.44	6.78 ^b^ ± 0.52	6.79 ^b^ ± 0.53	<0.001	0.805	0.763
**WHC-M/RZ**	0.57 ^b^ ± 0.10	0.56 ^b^ ± 0.06	0.70 ^a^ ± 0.12	0.63 ^ab^ ± 0.10	<0.001	0.190	0.396
**WHC-M/T**	0.36 ^b^ ± 0.04	0.36 ^b^ ± 0.02	0.41 ^a^ ± 0.04	0.39 ^ab^ ± 0.04	<0.001	0.232	0.413
**Cooking loss (%)**	38.34 ^a^ ± 1.75	33.93 ^b^ ± 1.46	37.17 ^a^ ± 1.83	33.30 ^b^ ± 2.24	0.099	<0.001	0.610
**WBSF (N)**	56.03 ^b^ ± 6.65	52.98 ^bc^ ± 3.96	61.02 ^a^ ± 6.76	50.13 ^c^ ± 5.34	0.526	<0.001	<0.001
**Color sensoric** **(1–8)**	4.50 ± 1.15	4.50 ± 0.60	4.50 ± 0.83	4.54 ± 0.58	0.930	0.930	0.930
**Marbling scores** **(1–7)**	4.08 ^a^ ± 1.00	3.25 ^b^ ± 0.45	3.00 ^b^ ± 0.74	4.17 ^a^ ± 0.83	0.713	0.464	<0.001

*L** = a measure of darkness/lightness (higher value indicates a lighter color); *a** = a measure of redness (higher value indicates a redder color); *b** = a measure of yellowness (higher value indicates a more yellow color); *C** = saturation index (higher values indicates greater saturation of red); *h* = hue angle (lower values indicates a redder color); *λ* = dominant wavelength; WHC-M = surface of the pressed meat film; WHC-T = surface of the wet area on the filter paper; WHC-RZ = WHC-T–WHC-M, a bigger WHC-M/T = ratio indicates a better WHC; CL = cooking loss; WBSF = Warner–Bratzler shear force; ^a,b,c^ Row means with different superscript differ in significance at *p* < 0.05.

**Table 4 animals-09-00941-t004:** Proximate composition (%) of fresh *M. longissimus lumborum* from investigated groups of Simmental cattle.

Parameter	1st Group (Same Herd)		2nd Group (Different Herds)		*p*-Values		
	Male	Female	Male	Female	Group	Gender	Group × Gender
**Moisture**	73.21 ^b^ ± 0.94	72.24 ^bc^ ± 0.99	74.54 ^a^ ± 1.32	72.11 ^c^ ± 1.43	0.086	<0.001	<0.001
**Protein**	21.32 ± 0.45	21.07 ± 0.72	21.18 ± 0.27	21.38 ± 0.68	0.625	0.868	0.165
**Total fat (IMF)**	4.38 ^ab^ ± 1.30	5.40 ^a^ ± 1.59	3.00 ^b^ ± 1.37	5.19 ^a^ ± 1.68	0.071	<0.001	0.185
**Total ash**	1.04 ^c^ ± 0.04	1.14 ^a^ ± 0.03	1.08 ^b^ ± 0.06	1.13 ^ab^ ± 0.07	0.254	<0.001	0.088

IMF = intramuscular fat. ^a,b,c^ Row means with different superscript differ in significance at *p* < 0.05.

**Table 5 animals-09-00941-t005:** Mineral composition (mg/100g) of fresh *M. longissimus lumborum* from investigated groups of Simmental cattle.

Parameter	1st Group (Same Herd)		2nd Group (Different Herds)		*p*-Values		
	Male	Female	Male	Female	Group	Gender	Group × Gender
**P**	152.28 ^a^ ± 14.92	106.91 ^b^ ± 5.37	157.97 ^a^ ± 8.15	110.26 ^b^ ± 11.96	0.152	<0.001	0.708
**Ca**	4.99 ^a^ ± 0.64	4.02 ^c^ ± 0.86	4.76 ^ab^ ± 0.73	4.23 ^bc^ ± 1.05	0.978	<0.001	0.366
**Na**	51.90^a^ ± 4.01	48.06 ^b^ ± 5.62	47.67 ^b^ ± 2.56	47.50 ^b^ ± 3.73	0.050	0.099	0.130
**Mg**	24.61 ^a^ ± 1.98	22.07 ^bc^ ± 1.33	21.03 ^c^ ± 3.20	23.78 ^ab^ ± 1.94	0.153	0.869	<0.001
**Fe**	1.89 ^b^ ± 0.18	2.09 ^b^ ± 0.36	1.91 ^b^ ± 0.24	2.46 ^a^ ± 0.34	<0.001	<0.001	<0.001
**Zn**	6.26 ^a^ ± 0.89	5.26 ^b^ ± 0.50	5.21 ^b^ ± 0.62	5.35 ^b^ ± 0.57	<0.001	<0.001	<0.001
**Cu**	0.03 ± 0.01	0.03 ± 0.01	0.03 ± 0.01	0.03 ± 0.01	0.921	0.370	0.728

P = phosphorus; Ca = calcium; Na = sodium; Mg = magnesium; Fe = iron; Zn = zinc; Cu = copper. ^a,b,c^ Row means with different superscript differ in significance at *p* < 0.05.

**Table 6 animals-09-00941-t006:** Fatty acid composition (g/100g fat) of fresh *M. longissimus lumborum* from investigated groups of Simmental cattle.

Parameter	1st Group (Same Herd)		2nd Group (Different Herds)		*p*-Values		
	Male	Female	Male	Female	Group	Gender	Group × Gender
**C14:0**	2.56 ^a^ ± 0.39	2.18 ^b^ ± 0.26	2.29 ^ab^ ± 0.29	2.54 ^ab^ ± 0.65	0.723	0.594	<0.001
**C16:0**	25.03 ^a^ ± 1.33	24.23 ^ab^ ± 1.18	23.32 ^b^ ± 1.68	25.36 ^a^ ± 1.43	0.483	0.137	<0.001
**C18:0**	19.28 ^a^ ± 1.92	16.37 ^c^ ± 1.69	18.50 ^ab^ ± 3.52	17.06 ^bc^ ± 1.52	0.944	<0.001	0.274
**C18:1**	42.21 ^bc^ ± 1.45	44.98 ^a^ ± 2.86	40.26 ^c^ ± 2.14	42.75 ^b^ ± 2.97	<0.001	<0.001	0.844
**C18:2**	4.01 ^a^ ± 0.37	3.09 ^b^ ± 0.70	4.08 ^a^ ± 0.53	3.13 ^b^ ± 0.41	0.722	<0.001	0.885
**∑SFAs**	46.88 ^a^ ± 1.59	42.68 ^b^ ± 1.91	44.11 ^b^ ± 4.39	44.00 ^b^ ± 2.10	0.362	<0.001	<0.001
**∑UFAs**	46.39 ^ab^ ± 1.33	47.83 ^a^ ± 2.36	44.84 ^b^ ± 1.55	45.38 ^b^ ± 3.04	<0.001	0.123	0.476
**∑OFAs**	6.74 ^b^ ± 1.75	9.49 ^a^ ± 3.41	11.05 ^a^ ± 3.98	10.62 ^a^ ± 3.57	<0.001	0.226	0.101

SFAs = saturated fatty acids (myristic acid—C14:0, palmitic acid—C16:0, stearic acid—C18:0); UFAs = unsaturated fatty acids (oleic acid—C18:1, linoleic acid—C18:2); OFAs = other fatty acid. ^a,b,c^ Row means with different superscript differ in significance at *p* < 0.05.

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
