# Peer review of "Influence of Farm Management for Calves on Growth Performance and Meat Quality Traits Duration Fattening of Simmental Bulls and Heifers"

_animals, 2019, doi:10.3390/ani9110941_

Round 1

Reviewer 1 Report

General comments

I believe that the study was well designed, conducted, and reported. My primary concern is have the authors stated the objectives for this research and the possible practical applications of it. The last paragraph of the Introduction (Lines 74 to 80), especially the last sentence in this section, report findings much like the study already was conducted. Some clarification and greater focus on why the study was conducted and what the impact of the findings will be would be very helpful.

There are areas of the manuscript where English grammar could be improved. It is hoped that if the manuscript is considered for acceptance that through the efforts of the authors and the editorial staff, these improvements can be made.

Specific comments

Line 14 – Recommend placing a comma after “nutrition”

Lines 30 to 31 – This sentence is difficult to understand. Please re-read and consider rewriting it. The problem word is “could” on line 30.

Line 36 – “…samples…were darkest…”

Lines 43 to 44 – Should this sentence read, “Bulls grow faster and more efficiently, have a higher slaughtering proportion, and produce leaner carcasses with a higher proportion of total meat than heifers.”?

Lines 45 to 46 – It would help to have this sentence rewritten more like the suggestion for the previous sentence. It is not incorrect as written, but stating it in the past tense with the citation at the end just reads awkwardly.

Line 199 – Need to add space after the period.

Line 209 – Recommend removing “has” here.

Line 287 and 293 (and probably elsewhere) – Beginning the sentence with the citation in brackets just reads awkwardly. Is there another way to do this so that maybe the authors and then the citation in brackets begins the sentence? On line 365, the sentence begins as “De Smet et al. (2000),” which reads much better than with starting the sentence with the citation in a bracket.

Author Response

Reviewer(s)' Comments to Author:

Review 1

General comments

I believe that the study was well designed, conducted, and reported. My primary concern is have the authors stated the objectives for this research and the possible practical applications of it. The last paragraph of the Introduction (Lines 74 to 80), especially the last sentence in this section, report findings much like the study already was conducted. Some clarification and greater focus on why the study was conducted and what the impact of the findings will be would be very helpful.

Thank You very much for acknowledging the significance of the Manuscript and for the suggested corrections. We have improved the Manuscript thoroughly and hopefully in its present form is ready to be considered for publishing.

There are areas of the manuscript where English grammar could be improved. It is hoped that if the manuscript is considered for acceptance that through the efforts of the authors and the editorial staff, these improvements can be made.

The English language, typos, grammatical errors throughout the Manuscript are revised.

Specific comments

Line 14 – Recommend placing a comma after “nutrition”

Thank You for the suggestion. It is revised now in the new version of the Manuscript.

Lines 30 to 31 – This sentence is difficult to understand. Please re-read and consider rewriting it. The problem word is “could” on line 30.

The phrase “could” is excluded from the Manuscript.

Line 36 – “…samples…were darkest…”

The sentence is now rewritten to be more precise.

Lines 43 to 44 – Should this sentence read, “Bulls grow faster and more efficiently, have a higher slaughtering proportion, and produce leaner carcasses with a higher proportion of total meat than heifers.”?

Thank You for the suggestion. It is revised now in the new version of the Manuscript.

Lines 45 to 46 – It would help to have this sentence rewritten more like the suggestion for the previous sentence. It is not incorrect as written, but stating it in the past tense with the citation at the end just reads awkwardly.

Thank You for the suggestion. It is revised now in the new version of the Manuscript.

Line 199 – Need to add space after the period.

Thank You for the suggestion. It is revised, as suggested.

Line 209 – Recommend removing “has” here.

It is removed, as suggested.

Line 287 and 293 (and probably elsewhere) – Beginning the sentence with the citation in brackets just reads awkwardly. Is there another way to do this so that maybe the authors and then the citation in brackets begins the sentence? On line 365, the sentence begins as “De Smet et al. (2000),” which reads much better than with starting the sentence with the citation in a bracket.

Authors appreciate comment and agrees that discussion is unclear and confuse. The corresponding disscussion with citation is more clear now.

Reviewer 2 Report

please check the enclosed file

Author Response

Review 2

Title: the title sounds confused and did not presents in clear and simple manner the experiment done.

Thank You for Your comment, the main purpose of the experiment is to determine the effect of identical paragenetic factors on different animal genotypes and their expression at the phenotypic level of the observed properties, so the title was derived accordingly to that.

Simple Summary: It appears confused and the language form should be strongly improved.

The English language, typos, grammatical errors throughout the Manuscript are revised.

Abstract: The organization of the abstract should be improved. The aim of the trial seems firstly to identify the effect of farm management before rearing on growth performance, but then authors described another aim of the trial which regards how the fattening systems can reduce the differences in growth performance due to different age and weight at rearing. Instead So abstract should be completely rewritten.

Thank You for the suggestion. Authors appreciate comment and agrees that discussion is unclear and confuse. The corresponding disscussion with citation is more clear now.

Introduction: authors exposed an adequate background concerning the topic of the trial, but it is not clear the link between the role of traceability that they described and cattle identification systems in beef production systems with the aim of the research. The aim of the trial is not exposed in a clear manner, it sounds that authors would have like expose as aim of the trial the effect of different rearing systems on growth parameters during the fattening period and slaughtering performance as well as meat quality traits but they concluded affirming that a period of fattening of around 400d is enough to eliminate differences among rearing system in the farms of origin: what about the aim of the trial? I suggest to consider a strong revision of introduction.

Thank You for the comment. We included some differencies in the Introduction according to your suggestions and it is revised now in the new version of the Manuscript.

Material and methods:

Animals and growth performance, slaughter procedure and carcass quality: the description of the feeding management before and after weaning is not clearly exposed, how is possible the use of a concentrate containing 25% of limestone flour? Moreover, if it is sufficiently clear the rearing management for calves in group 1, authors did not provided enough information about the calves in group 2 that come from different herds in term of rearing systems and feeding management. The age and weight at beginning of the trial expressed as mean±SD should be provided. The range of body weight at slaughter appears too wide. The diet during fattening period should be provided in a more precise manner and a table of its components expressed on % on DM basis and diet composition should be provided. Authors spoke about target slaughter age: it should be provided in terms of days or months.

I suppose that the reason for further consideration of the submitted Manuscript is possibly in an insufficiently clearly written method, and not about the wrong design of the experiment.

Slaughter procedure: it appears sufficiently clear, although the English form should be improved.

The English language, typos, grammatical errors throughout the Manuscript are revised.

Carcass quality traits evaluation: it was described in a clear manner.

Meat quality measurements: they are exposed with a sufficient clearness, concerning the meat fatty acid analysis, the extraction method appears as not the best for the actual standards and the paper used as reference (Van Oekel et al., 1999) refers to different methods for measuring water holding capacity and juiciness in pork and no information are showed about meat fatty acids extraction methods, authors should explain something about this. Moreover, concerning how the fat content in the meat (g/100 g of meat) and FAME content should be measured and data obtained analysed.

You are all right. It is clarified in the improved with right reference, in new version of the Manuscript.

Statistical analysis: it appears exposed in a clear manner.

Results and discussion: The main criticism of this section is linked to the criticism observed in the experimental plan and consequently to the results obtained, although I suggest the authors to consider that results achieved should be discussed justifying it explaining the possible biological effects, for example, of previous dietary management during in farm of origin rearing.

Growth performance: this section confirms the lack of a good experimental design: table 1 exposed the differences in terms of in vivo performance of the two experimental groups and interaction between sex and groups. Significant statistical difference had been found between groups and sex for slaughter age which was chosen as fixed parameter for both groups and sexes. Consequently, the results obtained are not comparable taking in account the experimental design presented.

Carcass quality traits evaluation: this section did not give more information respect those available in literature, two paper of three cited by authors did not concern slaughter performance in Simmenthal bulls, because one concerns the different mineral composition of meat from different beef breed and the other different strategies able to improve the beneficial fatty acids content in beef (Pilarcczyk, 2014; Scollan et al., 2006).

Thank you for your advice, unfortunately for this Manuscript could not be realized, but it would be the topic of our contiunated work.

Physical and sensory quality measurements: if discussion of results concerning colour parameters could be acceptable, some aspects regarding physical meat characteristics sounds not well presented as the discussion of results concerning WHC, cooking loss and marbling score for which I suggest to insert an explication of the differences between groups observed and not only a comparison with other researches.

Differences between Simmental bulls and heifers in relation to growth performance, carcasses and meat quality traits were significant and in agreement with our expectations that calves from the same herd have more similar final results and with those reported in the literature.

Proximate and Mineral composition: this section suffers of the same problems observed above, although for proximate analysis the effect of group is not significant. Concerning meat mineral content the effect of group was statistically significant for Fe and Zn, but authors did not justify the differences observed between the two experimental groups and more in general they limited the discussion only to comparison with results obtained by other authors.

Fatty acid profile: this section suffers of the same problems observed above and doesn’t take in

consideration the possible effects on meat fatty profile of the possible dietary effects during the preweaning period or during the permanence of animals in the farms of origin.

Reviewer 3 Report

Main comments

The manuscript requires corrections by the virtue of faults in the references and citing as well as due to methodological flaws. References are confused throughout the text (e.g. P3 L98 for [30], L138 for [44], P4 L148 [50] and L153 for [48], P6 L221 [39], P8 L291 [51] and L287 [53], and many others). Moreover, items from 55 to 59 are not cited in manuscript. It would seem that this manuscript was drawn up in haste, without proper attention.

It is surprising to note that authors have made use a lot of papers related to pig production and pork quality, it amounts to a grave misunderstanding. After all, the recent and subject-relative literature includes comprehensive and numerous articles related strictly with topic of presented study. Therefore I strongly recommend replacing irrelevant and excessive references [49, 50, 51, 52, 54, 55] with appropriate papers focused on beef. In general, authors should avoid excessive self-citations; this is particularly poor scientific writing. Moreover, reference must be made to the original source, not to a paper that has referred to these.

Section M&M should be supplemented by additional information about:

- how many farms (group 2) were included in experiment ?

- the precise location of muscle sampling should be given; provided by authors area between 6th and 7th rib rather points to thoracis section than lumborum; please correct here and hereafter

- ‘routine procedures’, whether it complies with stunning method or electrical stimulation? Please provide in details due to influence these factors on intrinsic properties as pH or colour of meat  

- description of statistical analysis is inconsistent with the results given in tables (there are provided main effects and interaction), therefore significant differences marked for mean values are misleading. In my opinion in tables the mean values for groups and sex should be supplemented, particularly, where the main effect is significant.

Other remarks

P2

L83: ‘on 48 calves of Simmental breed’

L84-85: use the plural form, here and hereafter (e.g. L89)

L85: ‘…system. They were weaned …’

P3

L95: provide an average value or range of body weight

L96: replace ‘food’ with ‘diet’

L114: replace ‘are’ with ‘were’

L127-135: please provide relevant reference (EC regulations) for EUROP grading of cattle carcasses

P4

L145: too many references, please limit the number to those that are methodologically valuable, similarly L146

L152: what was the correct thickness of samples (2.5 or 2.54 cm)?

L150-153: the term ‘sensory analyses’ is exaggerated; visual assessment included only colour and marbling evaluation   

L157-160: method of minerals determination should be detailed

L159: reference [45] refers to fatty acids determination

L165: reference [54] refers to WHC and juiciness determinations

P8

L283: replace ‘values’ with ‘percentages’ or ‘Cooking losses’

P10

L365: De Smet et al. - lack in Reference list

P11

L375: Is it the right reference?

Author Response

Review 3

Main comments

The manuscript requires corrections by the virtue of faults in the references and citing as well as due to methodological flaws. References are confused throughout the text (e.g. P3 L98 for [30], L138 for [44], P4 L148 [50] and L153 for [48], P6 L221 [39], P8 L291 [51] and L287 [53], and many others). Moreover, items from 55 to 59 are not cited in manuscript. It would seem that this manuscript was drawn up in haste, without proper attention.

Thank You very much for acknowledging the significance of the Manuscript and for the suggested corrections. We have improved the Manuscript thoroughly and hopefully in its present form is ready to be considered for publishing.

It is surprising to note that authors have made use a lot of papers related to pig production and pork quality, it amounts to a grave misunderstanding. After all, the recent and subject-relative literature includes comprehensive and numerous articles related strictly with topic of presented study. Therefore I strongly recommend replacing irrelevant and excessive references [49, 50, 51, 52, 54, 55] with appropriate papers focused on beef. In general, authors should avoid excessive self-citations; this is particularly poor scientific writing. Moreover, reference must be made to the original source, not to a paper that has referred to these.

Thank You for the comment. These citated references are the sources of methods which we used in this scientific work.

Section M&M should be supplemented by additional information about:

- how many farms (group 2) were included in experiment ?

Thank You for the question about number of farms which were included in experiment, according to that, we included the 5 different farms for the second group of cattle.

- the precise location of muscle sampling should be given; provided by authors area between 6thand 7th rib rather points to thoracis section than lumborum; please correct here and hereafter

Thank You for the suggestion, actually, M. longissimus lumborum (LL) was removed from the right side of each carcass, in the area between 6th and 7th rib to determine meat quality.

- ‘routine procedures’, whether it complies with stunning method or electrical stimulation? Please provide in details due to influence these factors on intrinsic properties as pH or colour of meat  

According to this question , with term routine procedures we meant about stunning method of cattle in slaughterhouse like the most used method.  

- description of statistical analysis is inconsistent with the results given in tables (there are provided main effects and interaction), therefore significant differences marked for mean values are misleading. In my opinion in tables the mean values for groups and sex should be supplemented, particularly, where the main effect is significant. 

All data are presented as mean and standard deviation and in accordance with that, further we presented them through the tables and discussion.

Other remarks

P2

L83: ‘on 48 calves of Simmental breed’

Thank You for the suggestion. It is revised now in the new version of the Manuscript.

L84-85: use the plural form, here and hereafter (e.g. L89)

Thank You for the suggestion. It is revised now in the new version of the Manuscript.

L85: ‘…system. They were weaned …’

The sentence is now rewritten to be more precise.

P3

L95: provide an average value or range of body weight

Thank You for the suggestion. It is provided now with a range of body weight for bulls and heifers.

L96: replace ‘food’ with ‘diet’

Thank You for the suggestion. It is replaced, as suggested

L114: replace ‘are’ with ‘were’

Thank You for the suggestion. It is corrected, as suggested in right tense.

L127-135: please provide relevant reference (EC regulations) for EUROP grading of cattle carcasses

Thank You for the suggestion. Now, it is provide with relevant reference (EC Regulations) for EUROP grading of cattle carcasses which we used in our work.

P4

L145: too many references, please limit the number to those that are methodologically valuable, similarly L146

Thank You for the suggestion. It is indicated now, at the improved version of the Manuscript.

L152: what was the correct thickness of samples (2.5 or 2.54 cm)?

Thank You for the question, the correct thickness of samples were 2.54 cm.

L150-153: the term ‘sensory analyses’ is exaggerated; visual assessment included only colour and marbling evaluation  

 Thank You for Your suggestion. You have right about the term sensory analyses could be  overstated for only two parameters of sensorical traits of meat, but in our case we thought that is relevant to the rest of parameters of technological-sensory traits.

L157-160: method of minerals determination should be detailed

It is detailed with relevant references in the part of Methods as you suggested.

L159: reference [45] refers to fatty acids determination

It is replaced with relevant reference, as suggested.

L165: reference [54] refers to WHC and juiciness determinations

Thank You for Your suggestion. It is replaced with relevant reference, as suggested.

P8

L283: replace ‘values’ with ‘percentages’ or ‘Cooking losses’

Thank You for Your suggestion. It is replaced with suggested term.

P10

L365: De Smet et al. - lack in Reference list

Thank You for the suggestion. It is provided now in Reference list of used literature.

P11

L375: Is it the right reference?

The right reference is now according to citation in that part of Manuscript.

Author Response

Review 4

General comment

Do not repeat in the text data already showed in the Tables. In addition, do not use the word significantly, just declare highest or lowest or what correspond in each case. Tables shows the p-values (or you mentioned in the text), so you do not need to use word significantly.  Some of the paragraphs are basically a repetition of the information already shown in the Tables. Authors should avoid doing that (i.e mineral composition, fatty acids composition etc.) They should summarize the most relevant information in the text, but not repeating it.

Thank You very much for acknowledging the significance of the Manuscript and for the suggested corrections. We have improved the Manuscript thoroughly and hopefully in its present form is ready to be considered for publishing. Also, thank You for Your insightful comment.

Specific comments

Line 12. enviroments instead of ambiance.

It is removed, as suggested.

Line 14 early life and improve

Thank You for the comment. It is included in The Manuscript.

Line 19. profits instead of remuneration

Thank You for the suggestion, it is replaced for the most suitable term as you recommended.

Line 24. This study assesd the effects of farm management during…

Thank You for the suggested. It is included in The Manuscript.

Line 25. meat quality traits during the fattening

It is inserted, as suggested.

Line 27. n=12 female). The 2nd group

Thank You, it is inserted, as suggested.

Line 28-29. Calves were transferred to a feedlot and fed with a commercial feedlot ration at three to four months of age.

Thank You for the suggested version of sentence, it is more clear and is inserted.

Line 30. The aim was to determine if identical fattening conditions

It is inserted in the new version of Manuscript.

Line 45-46. Therefore, the meat from heifers compared to bulls have more dry matter and intramuscular fat, and is more tender and acceptable

Thank You for the suggestion. It is revised now in the new version of the Manuscript.

Line 48. Further, it has been shown that

Thank You for the suggestion. It is revised now in the new version of the Manuscript.

Line 66. What the authors means by “and return protein purchasing”?

Thank You for the question. It is excluded from the Manuscript.

Line 78. Delete samples

Thank You, it is inserted, as suggested.

Line 79. Authors must be clear about what they expect in their hypothesis statement. Indication of that there will be differences is not good enough. Which group will be better? What type of of influence? positive or negative?

Thank You for the question. It is revised now in the new version of the Manuscript, we indicate that the group which came from the same herd will be more similar at final results unlike the group from different herds.

Line 84. A total of 24….

Thank You, it is inserted.

Line 88. came instead of were

It is inserted, as suggested.

Line 92. age when transferred

Thank You for the suggestion, it is inserted, as suggested.

Line 95. It It could be better if indicate weight ranges for bulls and heifers. Because the range is too wide and that can affect meat quality traits.

Thank You for the suggestion. It is provided now with a range of body weight for bulls and heifers.

Line 101. Cattle were fed

Thank You for the suggestion.

Line 106. Delete which

Thank You for the suggestion, it is done.

Line 107-108. Individual calves weights were measured using a heavy duty scale with accuracy ± 0.5 kg (initial weight) at the beginning and the end of the fattening period prior to slaughter

It is corrected, as suggested in the improved version of the Manuscript.

Line 116. All the cattle were slaughtered according to routine procedures of the slaughterhouse

Thank you for the suggestion, it is inserted in the Manuscript.

Line 144. described by Senk et al [44].

Thank you for the suggestion, it is inserted in the Manuscript.

Line 150. as described by Senk et al [44].

Thank you for the suggestion, it is inserted in the Manuscript.

Line 162. Meat samples of 5 g were dried… Then samples were…

Thank You for the suggestion. It is revised now in the new version of the Manuscript.

Line 175. If possible presente mean and standard error of the mean.

Line 176. ANOVA (gender and group) within

Thank you for the suggestion, it is inserted in the Manuscript.

Line 180. Why the interaction group x gender was not included in the model? (After review Table 1, there was interaction, but it was not declared in the statistical model)

Thank you for the question, you have right about that, as suggested, now it is in the improved version of the Manuscript.

Line 186 Delete all the line

It's done in the new version of the Manuscript.

Line 192. which is characterized by fat deposition during the “finishing” period [1]. (In this context, it is not clear if the criteria to end the experiment was the live weight or the fat deposition. It seems that was the second, because male of 2nd group reached a higher slaughter weight, that is +45 kg).

Thank that You pointed to ambiguity. That sentence was now more clear according to the criteria of the fat deposition.

Line 197. females population

That is inserted in the text, as you recommended.

Line 198-204. There was interaction bewteen Gender and Group for total gain during the fattening period (P<0.001), with bulls achieving higher total gain than heifers. In our study, heifers from 2nd group achieved lowest total gains and slaughter weight (383 kg and 518 kg) during the fattening period in comparison with the others animals. Likewise, heifers from the 2nd group spent the longest period at the feedlot (456.8 days) which corresponded to before mentioned claim that calves from different herds with different rearing practices should take longer period to adapt.

Thank You for the suggestions about sentences. It is revised now in the new version of the Manuscript.

Line 205. Growth performance per groups and gender of Simmental cattle fattened in a feedlot.

That is inserted in the text.

Line 210. Similarly, Bureš et al., [6] found

Thank You for the suggestion. It is revised now in the new version of the Manuscript.

Line 211-214. Furthermore, gender and interaction of group with gender had the significant influence on the slaughter weight of cattle. We indicated statistically significant differences between cattle where bulls were significantly heavier (568 and 613 kg) than heifers (547 213 and 517 kg) at 1st and 2nd group, respectively, prior to slaughter.

Thank that You pointed to ambiguity. That sentence was omitted from the text.

Line 217. The carcass quality traits evaluation for investigated groups of cattle are shown in Table 2.

Thank You for the suggestion. It is revised now in the new version of the Manuscript.

Line 218. Hot carcass weight was dependent on gender and group. Gender and interaction between gender and group had significant influence on hot carcass weight.

Thank that You pointed to ambiguity. That sentence was omitted from the text.

Line 219. Hot carcass…. (delete Thus)

Thank you for the suggestion, it is deleted from the Manuscript.

Line 220. in 1st and 2nd group, respectively ( P < 0.001).

That is inserted in the text, as you recommended.

Line 224. Also, Herva et al., [13] concluded …

Thank you for the suggestion, it is inserted in the Manuscript.

Line 232. traits evaluation (P > 0.27). Kaminiecki et al., [23] found….

Thank you for the suggestion, it is inserted in the Manuscript.

Line 235. in a significantly

Thank You for the suggestion.

Line 236. Fat cover scores were significantly

Thank you for the suggestion, it is deleted from the Manuscript.

Line 239. Usually females start to deposit fat earlier than males. In addition, the males were intact (with their testicles), so they should be leaner than heifers.

Thank You for the comment. It is revised now in the new version of the Manuscript.

Line 240. by Chambaz et al., [8] for

Thank You for the suggestion. It is revised now in the new version of the Manuscript.

Line 241. According to Kukolj et al., [25] irrespective…

Thank You for the suggestion, it is done.

Line 250. by Pilarcyk [39]

Thank You for the suggestion. It is revised now in the new version of the Manuscript.

Line 253. taste [53]. We found an interaction effect between

Thank You for the suggestion. It is revised now in the new version of the Manuscript.

Line 255. When authorts mention cattle are they referring to gender or group?

Thank You for the question. In this part of the Manuscript  (regarding the carcass properties, actually conformation) we reffered to all of the cattle in general, because the majority of carcasses were classified as class R.

Line 270. as well [53]. Concomitant (space after the period)

That part is include in the text

Line 284. Scollan et al., [43]

Thank You for the suggestion. It is revised now in the new version of the Manuscript.

Line 287. The USDA [53]

That part was replaced in the text with right reference [58].

Line 289. by Bureš et al., [7]

Thank You for the suggestion. It is revised now in the new version of the Manuscript.

Line 290. and for Simmental bulls (48.19 N) [42].

Line 291. has a lower

Thank You for the suggestion. It is revised now in the new version of the Manuscript.

Line 293. Scollan et al., [43] demonstrated

That part is include in the text.

Line 295-297. There was an interaction effect between group and gender for marbling score. Marbling score was significantly highest (P<0.05) for the heifers at 2nd group (4.17) than the others animals.

It is corrected, as suggested in the improved version of the Manuscript.

Line 299-305. The proximate composition of meat samples from Simmental cattle are shown in Table 4. We found an interaction effect between group and gender for moisture content (P<0.001), where the bulls had higher moisture contents (73.21 to 74.54%) than heifers 301 (up 72.11 to 72.24%) for 1st and 2nd group, respectively. Proximate composition, except protein was influenced mainly by gender. No differences were found in the content of protein among meat samples from two groups. As expected, the protein content from was in agreement with some earlier investigations [39, 43].

The sentence is now rewritten to be more precise.

Line 307-308. as was 307 expected.

That part was omitted from the text.

Line 309. the USDA [53]

That part was replaced in the text with right reference [58].

Line 318. Gender affected the content of phosphorus,

That is inserted in the text.

Line 342-359. Do not repeat the values, uses for example percentages to explain which fatty acids were the most importants.

Thank you for the suggestion, the sentence is now rewritten to be more precise.

Line 377. of above 400. (Authors must be cautious with this generalization because animals with early under nutrition (prenatal or postnatal) may not fully recover)

Thank you for the suggestion, the sentence is now rewritten to be more precise.

Line 379-390. I suggest reviewing the conclusions, currently they are extensive. Just remember that they must be supported by the results showed in the manuscript and give response to the hypothesis.

Authors appreciate comment and agrees that discussion is unclear and confuse. The corresponding disscussion with citation is more clear now.